# Red Code Management in a Pediatric Emergency Department: A Retrospective Study

**DOI:** 10.3390/children11040462

**Published:** 2024-04-12

**Authors:** Serena Bertone, Marco Denina, Manuela Pagano, Angelo Giovanni Delmonaco, Emanuele Castagno, Claudia Bondone

**Affiliations:** 1Paediatric Unit, Ospedale Regina Montis Regalis, 12084 Mondovì, Italy; serena.bertone@unito.it; 2Paediatric Infectious Diseases Unit, Regina Margherita Children’s Hospital, University of Turin, Città della Salute e della Scienza, 10126 Turin, Italy; 3Department of Pediatric Emergency, Regina Margherita Children’s Hospital, Città della Salute e della Scienza, 10126 Turin, Italy; mpagano@cittadellasalute.to.it (M.P.); adelmonaco@cittadellasalute.to.it (A.G.D.); ecastagno@cittadellasalute.to.it (E.C.); claudiabondone@virgilio.it (C.B.)

**Keywords:** red code, emergency department, diagnostic pathways

## Abstract

The “red code” (RC) represents the highest level of emergency in the emergency department (ED). This study retrospectively analyzed RCs in the Regina Margherita Children’s Hospital ED, a regional referral center in north Italy, between 1 July 2020 and 30 June 2023. The aim was to describe RC characteristics and to identify significant correlations between presenting complaints and clinical management. The study includes 934 RCs (0.9% of overall ED admissions); 64% were assigned based on the Pediatric Assessment Triangle alteration. Most patients, 86.5%, followed the medical pathway, while 13.5% were surgical cases. Admission complaints were respiratory (46.9%), neuropsychiatric (26.7%), traumatic (11.8%), cardiologic (9.3%), metabolic (3.8%), and surgical (1.5%). Seventy-six percent of patients received vascular access, and intraosseous access was obtained in 2.2% of them. In one-third of RCs, an urgent critical care evaluation was necessary, and 19% of cases required admission to the intensive care unit. The overall mortality rate was 3.4% (0.4% in ED setting). The study identified six distinct diagnostic pathways, each associated with specific characteristics in clinical presentation, management, therapeutic interventions, and outcomes. Our findings underscore the need for a systematic approach in pediatric emergency settings, supported by international and national guidelines but also by clearly defined diagnostic pathways, aiming to enhance the quality of care and patient outcomes.

## 1. Introduction

In the emergency department (ED), an adequate triage system is needed to ensure that children with severe conditions can be immediately identified and quickly directed toward appropriate care [1,2,3]. The red code (RC) represents the highest urgency and requires immediate medical assessment [4]. In the international literature, there is a scarcity of epidemiological studies related to RCs ED admissions in the pediatric age group [4,5,6]. The goal of this study was to retrospectively describe epidemiological characteristics, admission features, clinical management, and the outcome of RCs in a tertiary urban teaching children’s hospital, aiming to provide a systematic approach in a pediatric emergency setting.

### 1.1. Background

#### 1.1.1. Triage

Triage is an important process for prioritizing patients according to their clinical needs, usually managed by trained nurses. There are various triage systems implemented around the world; the universal goal is to supply effective care to patients while optimizing resource usage and timing. In recent years, a five-level triage system has been introduced in pediatric ED, identified by numbers or colors (1—red, 2—orange, 3—blue, 4—green, 5—white) corresponding to a decreasing degree of urgency [7]. The triage code assignment derives from different evaluations. The first component is the “quick look”, a rapid assessment that does not require equipment but only visual and auditory evaluation to assess the appearance of the patient, respiratory function, and skin circulation, reflecting the brain function, oxygenation, ventilation, and perfusion of the child [1,8,9,10], followed by the evaluation of the main complaint and the vital signs, including pain. The RC is defined as the absence or compromission of one or more vital functions and represents the priority access to medical evaluation; it constitutes a minority among pediatric ED admissions, representing 0.25–1% of total admissions according to local statistics [4,8].

Table 1 shows a description of the five-level triage system.

#### 1.1.2. RCs Management

The routine clinical management of RCs is based on the Pediatric Advanced Life Support (PALS) approach, a systematic method for managing children in critical condition [9]. Alongside the PALS algorithm, international guidelines and local hospital protocols aim to standardize care and optimize outcomes [10,11,12,13]. Diagnostic assessments for detecting and identifying severe clinical conditions include bedside, laboratory, and instrumental tests [9,14].

In recent years, increasing importance has been ascribed to point-of-care ultrasound (POCUS), closely related to the operator’s experience but extremely useful in emergency settings as a rapid, reproducible, portable, and non-invasive method. The main scenery of its application is polytrauma by the e-FAST protocol [15].

## 2. Materials and Methods

We retrospectively analyzed all the patients admitted as RCs to the Regina Margherita Children’s Hospital ED, a regional referral center in North-Western Italy, between 1 July 2020 and 30 June 2023.

### 2.1. Hospital and Emergency Department Organization

Regina Margherita Children’s Hospital is a referral pediatric center particularly for the Piedmont and Aosta Valley regions. It has surgical, medical, and diagnostic specialties for the cure of neonates, infants, children, and adolescents for rare, chronic, and complex diseases. The hospital provides a pediatric intensive care unit (PICU) managed by an anesthesiologic team (10 beds) and a total of 215 beds in pediatric general and specialistic wards.

The ED is among the largest in Northern Italy in terms of patient visits, with a case load of 40,000 patients treated per year. It serves as a reference for pediatric emergencies across multiple specialties (including cardiac surgery, neurosurgery, neonatal surgery, and burns) in the Piedmont and Aosta Valley regions, and it is the site of a Pediatric Trauma Center.

The age range for ED admission is 0–14 years, extended to 18 years for patients with chronic diseases followed up at the hospital. Triage based on the five-level system is managed by trained nurses. In our ED, there is a distinction between “medical” and “surgical” patients, with separate pathways, waiting times, and healthcare staff. Our ED has a related ward of a short observation unit (10 beds), lasting up to 36 h.

In our ED point-of-care (POC), blood tests are available, including a blood count with c-reactive protein and multiparametric blood gas (MBG, including glucose, hemoglobin, creatinine, potassium, sodium, calcium ionized, chloride, lactate), useful for directing bedside clinical management.

### 2.2. Data Collection

We collected data about the epidemiology, ED admission characteristics, initial management, blood and instrumental tests, initial therapeutic approaches, and outcomes (short observation unit, pediatric ward or PICU admission, death) of the study population using computerized medical records of the hospital system.

We classified RCs admissions into six specific complaint categories: respiratory, cardiologic, metabolic, neuropsychiatric, surgical, and traumatic.

Among the data collected, there was a percentage of “not reported” ones, either because they were not detected or not recorded in the clinical records.

### 2.3. Red Code Definition

According to national guidelines, in our ED, the RC represents the priority access to medical evaluation, and it is defined as the absence or compromission of one or more vital functions. In particular, we have assigned the RC to patients with an alteration in one of the vital signs (heart rate HR, respiratory rate RR, oxygen saturation satO_2_) or in one of the parameters constituting the Pediatric Assessment Triangle (PAT, consciousness, breathing, and skin circulation)—for example, a patient who is unconscious or cyanotic [8,10,16].

### 2.4. Pathologic Items Definition

Table 2 shows the criteria we used to define the pathologic items in our study.

### 2.5. Statistical Analysis

The statistical analysis was performed using IBM SPSS Statistics 27.0 (IBM Corp., Armonk, NY, USA). Significance was set at *p* < 0.05. All *p*-values were two-tailed. In the descriptive analysis, categorical variables are reported as absolute numbers and percentages, and continuous variables are reported as the mean, median, and interquartile range (IQR), as appropriate. To compare the continuous variables of the study groups, a Student’s *t*-test was used. To evaluate the discrete variables, Pearson χ^2^ and correlation Fisher exact tests were performed, as appropriate.

### 2.6. Strenghts and Limits

The strengths of our analysis included the large size of the study population and the setting in which it was conducted: being a tertiary hospital, it was adequate for managing high-complexity codes. Among the study’s limits, there was a percentage of “not reported” data, either because they were not detected or not recorded in the clinical records. Since this was a retrospective study, it would be suitable to expand it with a prospective study for a more comprehensive evaluation.

## 3. Results

### 3.1. Population Study

A total of 934 patients triaged as RCs were admitted to our ED between 1 July 2020 and 30 June 2023, equal to 0.9% of 105.798 total ED admissions in the period of the analysis. A total of 55.9% were male, and the median age was 3.2 years (1.1–7.4). The medical pathway was prevalent (808 patients, 86.5%), while the surgical pathway consisted of 126 patients (13.5%). The different sex distribution between medical and surgical patients was statistically significant, with 54.2% of males in the medical group and 66.7% in the surgical one. Comorbidities were detected in 344 medical patients (42.6%) and in 11 surgical ones (8.7%) (*p* < 0.05). A total of 50% of the medical patients (404) arrived at the ED independently, while 65.1% of the surgical patients (82) were transported by ambulance (*p* < 0.05). The main complaints were: respiratory (438 patients, 46.9%), neuropsychiatric (249 patients, 26.7%), traumatic (110 patients, 11.8%), cardiologic (87 patients, 9.3%), metabolic (36 patients, 3.8%), and surgical (14 patients, 1.5%). The majority of the patients (64%) were triaged as RC due to PAT alteration, which was the main finding for RC definition in 100% of surgical patients. Table 3 shows the general descriptive characteristics of the total study population according to the medical and surgical pathways.

### 3.2. ED Admission Vital Signs

Among ED admission vital signs, the level of consciousness, modality of breathing, body temperature, and skin description were registered in the majority of patients, both medical and surgical. Blood pressures and capillary refill times were detected in a minority of cases. In detail, the AVPU and breathing description were reported in all 808 medical patients (100%) and in 125 surgical ones (99.2%); skin was described in 765 medical patients (94.7%) and in 106 surgical ones (84.1%); blood pressure was measured in 135 medical patients (16%) and in 66 surgical patients (52.4%), with pathologic values in 3.2% and 10.3% of cases, respectively; the capillary refill time was reported in 284 medical patients (35.1%) and in 13 surgical ones (10.3%), with pathologic detection in respectively 7.4% and 3.2% of cases. The difference found in vital signs at ED admission between medical and surgical patients was statistically significant for all parameters analyzed. Table 4 shows the vital signs at ED admission of the study population according to the medical and surgical pathways.

### 3.3. Vascular Access

Vascular access, including both pre-hospital and ED cannulation, was provided in 714 patients (76% of the total study population): 600 were medical (84%) and 114 were surgical (16%). This means that venous access was provided in 74% of medical patients and in 90% of surgical ones. Instead, 202 medical patients (25.1%) and 5 surgical ones (4%) were managed without venous access in the ED setting. The different use of vascular access in the two pathways was statistically significant (Table 5).

The peripheral venous catheter (PVC) was the most used type of vascular access, placed in 707 patients (75.7% of the total study population). CVC (central venous catheter) was provided in 1 patient (1.1%) and intra osseous access (IO) was provided in 21 patients (2.2%); 15 patients (1.6%) were placed on both PVC and IO. In detail, among 21 patients managed with IO access, the main complaint was cardiologic in 8 patients (38%), neuropsychiatric in 5 (24%), respiratory in 4 (19%), and traumatic in 4 (19%); 12 of them (57%) were admitted to PICU, and death occurred in 9 of them (43%). These last two frequencies were higher than those of the total study population, as presented in the following paragraphs.

### 3.4. Emergency Tests

Blood count and multiparametric blood gas analysis were the main emergency tests performed. The use of these tests was different (*p* < 0.05) between medical and surgical patients: blood count was performed in 554 medical patients (68.6% of the medical group) and in 98 surgical patients (77.8% of the surgical group), while multiparametric blood gas analysis was performed in 636 medical patients (78.7% of the medical group) and in 73 surgical ones (57.9% of the surgical group). A significant difference between medical and surgical pathways was also found in the performance of ECG and CT scan: ECG was performed in a selected part of patients (19% of total patients) and CT scan was mostly used in surgical RCs (10.1% of medical patients vs. 64.3% of surgical ones). US was executed in about one-third of patients (including both POCUS and specialistic US), without a significant difference between the two pathways. The anesthesiologist was involved in 54% of surgical RCs but only in one-quarter (25.5%) of medical ones, with a significant difference between the two pathways. Table 5 shows the clinical management of the medical and surgical populations.

### 3.5. Therapeutic Approach

Table 6 shows a detailed breakdown of drugs’ macro-categories administration in the study population according to the medical and surgical pathways.

In detail, 548 total patients (58.7%) were supported by oxygen or ventilation: 521 of them belonged to the medical pathway (equal to 64.5% of this group) and 27 belonged to the surgical one (equal to 21.4% of this group), with a significant difference between the two pathways. A significant difference was also found for drugs acting on airways, which were used in 40.2% of total patients, represented exclusively by medical patients. Antibiotics or antivirals were administered to 18.4% of patients (without a significant difference between the two pathways), while steroids, antipyretics, and analgesic were administered to about one-third of the study population. Diuretics and psychoactive drugs were exclusively used in medical patients, in 1.5% and 1.4% of them, respectively. Dressings were realized in 2.6% of total patients and they were mainly used in surgical patients (18.3% of this patient group, with a significant difference compared to the medical group). Electric therapy has never been used.

### 3.6. Mortality Rate

The mortality rate of RCs in the ED setting was 0.4% (four medical patients). The overall mortality rate was 3.4% (32 patients of the total study population), presenting in 2.8% of medical patients (23 patients) and in 7.1% of surgical ones (9 patients), with a statistically significant difference between the two pathways. While 12.5% of total deaths occurred in ED, PICU was the setting where death mainly occurred (24 patients, equal to 75% of deaths), and the remaining 12.5% (4 patients) died in hospital wards.

### 3.7. Hospital Admission

A total of 68 patients (7%) were directly discharged from ED. Instead, regarding hospital admission after ED management, 175 patients (18.7%) were admitted to PICU, 274 (29.3%) were admitted to a short observation unit, and 410 (44%) were admitted directly to hospital wards. PICU admission has been the choice for 120 medical patients (14.9%) and for 55 surgical ones (43.7%), with a significant difference between the two pathways. A significant difference between the two pathways was also found in the admission to the short observation unit, chosen for 269 medical patients (33.3%) and for only 5 surgical ones (4%). In the overall management of hospital admissions, 702 patients (75.2%) were admitted to hospital wards or directly from ED, following PICU, or after a short observation unit. Table 7 shows the hospital admission modality of the total study population according to the medical and surgical pathways.

The six main complaints were characterized by different durations of hospital admission, calculated from the ED triage until final discharge, including recovery in the short observation unit, PICU, and hospital ward. The mean duration of hospitalization is illustrated in Table 8.

### 3.8. Diagnostic Pathways

In our study population, we identified six clearly different diagnostic pathways, each one defined by specific characteristics in term of admission parameters, clinical management, therapeutic approaches, and outcomes: respiratory, neuropsychiatric, traumatic, cardiologic, metabolic, and surgical. The cardiologic category included cardiac arrhythmias and circulatory disturbance with all types of shock; the metabolic group primarily consisted of glycemic abnormalities, both severe hypoglycemia (for example, in children with congenital metabolic diseases) and ketoacidosis at the onset of type 1 diabetes; the smallest pathway (1.5%) consisted of true surgical patients, represented by cases of acute abdomen or neurosurgical emergencies.

Regarding the RC definition, 100% of metabolic, neuropsychiatric, and traumatic patients were triaged as RCs for PAT alterations, while 69% of respiratory patients were triaged for desaturation at admission. Comorbidities were associated with neuropsychiatric complaints in 50% of cases (including both neurologic diseases and psychiatric disturbances) and they were absent in 93% of traumatic RCs. Among admission vital signs, 86% of respiratory patients were alert, 24% were supported by oxygen, and 74% were with normal skin; in contrast, among traumatic RCs, 31% were unresponsive, 28% were intubated, and 72% had pathologic skin; AVPU was pathologic (P or U) in 59% of neuropsychiatric patients; altered skin was found in cardiologic patients (paleness, cyanosis, marbling) and surgical ones (burn, ecchymosis, wound, abrasion). Regarding clinical management, a blood count was performed in 84% of traumatic RCs; multiparametric blood gas analyses were performed in 78% of respiratory and in 97% of metabolic ones; electrocardiograms were performed in 26% of neuropsychiatric, 58% of metabolic, and 52% of cardiologic patients. A total of 53% of traumatic RCs requested anesthesiologic consultation and 43% requested PICU admission, both rare in respiratory emergencies (23% and 15%, respectively). Deaths were statistically significantly associated with cardiac complaints (13% of these patients) and traumatic complaints (7%), while the mortality rate among respiratory patients was lower than statistically expected (1.5%). Table 9 represents in detail the statistically significant correlations between diagnostic pathways and ED management items.

## 4. Discussion

The international literature is lacking in epidemiological studies related to RCs in pediatrics, and these studies are often characterized by very different socio-cultural and environmental contexts.

The RCs constituted a minority of our ED admissions, representing less than 1% of total admissions; this low incidence is in line with local statistics and with data of similar studies conducted at the Children’s Hospital ED of Padua (1.276 RDs in 5 years) and at the Children’s Hospital ED of Trieste (251 RDs in 4 years) [4,6,8].

The medical pathway was the prevalent path and was characterized by a frequent presence of comorbidities related to the main complaint. This might be justified by the fact that our ED is part of a tertiary Children’s Hospital, where patients are followed for their underlying chronic condition but where they also seek care for acute issues. In our analysis, the respiratory problems were the most common ED admission complaint, and they were typically represented by respiratory failure primarily due to bronchitis, pneumonia, bronchiolitis, or an acute asthma attack in most cases. Regarding the access mode to the ED, most of the surgical patients were taken by ambulance, including both advanced transport with helicopter rescue and a territorial rescue system, used in particular for victims of accidents, in accordance with the international literature [17,18].

RCs require prompt recognition to ensure their adequate management and to improve their outcome [1,2,3]. According to the international literature, an efficient triage system is needed to a correct priority code assignment: alongside vital signs, nurses’ experiences are crucial to catching PAT alterations, which, in our study, were the main motives of the RC definition [1,3,19]. The assessment of vital signs plays a crucial role non only in triage code assignment but also in the management according to the PALS protocols, with the goal of the fast recognition of respiratory distress, respiratory failure, and shock to immediately take life-saving interventions [3,9,20]. We detected significantly different vital signs at ED admission between medical and surgical patients: this last group, having more compromised breathing and consciousness, more frequently required the intervention of the anesthesiologist. The blood pressure and capillary refill time were reported in a minority of cases, but these two parameters should be remembered as part of the primary assessment of critical patients for assessing peripheral perfusion [9].

Obtaining venous access in critically ill children is an essential procedure for restoring blood volume and administering drugs during pediatric emergencies. The first option for vascular access is through a peripheral vein puncture, but if this cannot be used or if it takes too long to be placed, the intraosseous route is an effective option for rapid and safe venous access [9,21]. In our analysis, vascular access was established in most patients: as expected, PVC was the most used type, while IO access was obtained in 2.2% of the total population. The international literature reports the rare use of intraosseous access in pediatric-aged patients, typically in extremely unstable and critical patients; however, the incidence varies significantly from one study to another (from 0.02% to 20%), according to the study population analyzed [5,22,23,24]. In our setting, typical clinical conditions managed with IO access were polytrauma, status epilepticus, severe respiratory insufficiency, and cardiocirculatory instability, all characterized by an extremely severe clinical picture: this also justifies the high rate of PICU admissions and deaths found among this category of patients.

In the ED setting, clinicians need to make accurate and timely decisions regarding emergency management, and POC tests have the potential to provide rapid and accurate results [9,14]. In our analysis, blood counts and multiparametric blood gas analysis were largely performed, and they provided a great part of pathologic results, confirming the utility of their execution. Electrocardiograms were significantly correlated not only with cardiologic complaints but also with metabolic and neuropsychiatric problems, in which it was necessary for finding anomalies induced by electrolyte disturbances and for calculating the corrected QT interval.

In recent years, we have assisted in the widespread adoption of POCUS in pediatric EDs; the international literature has emphasized its usefulness in the pediatric emergency setting, representing a non-invasive and rapidly performed technique, performable even on unstable patients during resuscitative maneuvers and useful not only in the diagnostic process but also in the primary assessment [9,15,25]. Unexpectedly, in our study, ultrasound was performed in only one-third of the patients, considering both bedside and specialistic exams: this could be, in part, attributed to the limited use of e-FAST US in polytrauma, which was more commonly managed through the execution of a CT scan, which provided pathological findings in a great portion of cases. The most frequently used US was the pleuropulmonary one, but we should implement the use of US as a component of the patient’s primary assessment, especially in cases of hemodynamic instability, as it allows for the rapid evaluation of cardiac function and volume status, in accordance with ongoing international literature [3,26,27,28].

Drugs used in the ED therapeutic approach can be divided into macro-categories [6] and we found some significative correlations between their administration in medical and surgical pathways, except for antibiotics, antiviral, antipyretics, and analgesic, which were used across different types of patients without specificity to any diagnostic pathway. Antiemetics, diuretics, psychoactive, and cardioactive drugs have been rarely used, and this may have affected their lack of statistical significance. Electric therapy, including both defibrillation and cardioversion, was never utilized and this reinforces the possibility that pediatric emergency medicine faculty are at a significant risk for skill deterioration, especially concerning critical procedures performed in a pediatric ED [24]. Statistically significant correlations were also found between the administration of some drugs and the admission main complaint: for example, oxygen, steroids, drugs acting on airways (mainly a bronchodilator via aerosol or intravenous administration), and antibiotics in respiratory patients; antiemetics, anticonvulsants, and psychoactive drugs (antipsychotics and benzodiazepines) in neuropsychiatric patients; crystalloids, colloids, sedatives, and dressings in traumatic patients; colloids, crystalloids, and cardioactive drugs (adenosine, adrenaline, atropine, beta-blockers) in cardiologic patients; crystalloids, glucagon, and insulin in metabolic patients; and dressings in surgical patients, all in accordance with the guidelines and the hospital protocols [9,11,12,13,29,30].

As supported by the literature, close working relationships between the ED and PICU are fundamental to ensuring better outcomes [31,32]. Pediatric settings are characterized by pediatricians as the frontline figures in facing emergencies, tasked with managing various consultants, including the anesthesiologist, unlike in the adult world, where RCs are often handled by specialists. In our setting, a low percentage of RCs were admitted to the PICU. The anesthesiologist was largely involved in surgical RCs because they were often admitted intubated (so they required advanced airway management) or in critical condition, requiring rapid management in the operating room. For these same reasons, approximately half of them were admitted to the PICU. Among surgical patients, the main part was represented by traumatic complaints, including polytrauma and burns: their critical condition justifies a higher PICU admission and a higher mortality rate than expected, with statistical significance. As mentioned earlier, PICU admission also had a statistically significant correlation with patients managed with intraosseous access.

Regarding the hospital admission modality, we usually reserved a short observation unit to RCs with a prognosis of rapid resolution (for example, febrile seizure or acute asthma attack), in accordance with the literature [33]. In our setting, we have also detected that, during the epidemic season, a short observation unit served as a temporary arrangement while ones awaits admission to a regular inpatient ward. The duration of hospitalization was extremely variable according to the main complaint, ranging from respiratory patients discharged directly from ED to psychiatric patients hospitalized for months.

The mortality rate of RCs in our ED setting was 0.4%, and the overall mortality rate was 3.4%. As expected, the overall mortality is lower compared to that in resource-limited countries but surprisingly higher compared to that of the Children’s Hospital of Padua (0.7%) [4,5,34,35]. This can be the consequence of a different organization of regional networks of trauma centers but also of the fact that Regina Margherita Children’s Hospital is a reference center for rare and complex conditions such as metabolic, neurosurgical, and cardiovascular diseases. As mentioned earlier, we found a higher mortality rate in patients managed with intraosseous access but also in patients with traumatic and cardiac complaints, representing the most critical patients’ categories. Regarding the setting where deaths occurred, in our study, only four patients died in ED, while PICU was the setting where death mainly occurred, partially related to underlying incurable comorbidities. Despite death being a rare event in the ED setting, its management underscores the importance of not only medical expertise but also the ability to provide compassionate and empathetic care during an incredibly distressing time, providing emotional support and comfort to the grieving family and ensuring adequate debriefing for the entire teamwork [36].

The RCs represent a wide heterogeneous category of patients united by critical conditions, but we identified six clearly defined diagnostic pathways. This reinforced the concept that RCs management should be based on a systematic approach, driven not only by the application of international protocols but also by a clear categorization of patients upon ED admission, allowing for specific management pathways and tailoring care. Regarding the respiratory pathway, which was the most common, a higher proportion than expected was managed without vascular access, without blood tests (performed in ED settings), and without anesthesiologic consultation, while deaths and PICU admissions were lower than predicted; a greater use was reserved for a short observation unit, often sufficient for solving the acuteness. These data might attest to the confidence of our ED team in managing respiratory RCs, supported by clinical experience and clear protocols [29,30], but a prospective analysis will be useful in confirming it. On the contrary, our ED was just a transitional point for a great number of surgical and traumatic patients, as they were directly managed in the PICU or specialized wards.

A systematic clinical approach supported by a defined diagnostic pathway enables action in the emergency setting according to standardized and defined protocols, reducing the risk of errors and improving outcomes. Clearly defined diagnostic pathways could also be a useful support for first-level pediatric EDs, where the daily number of admissions is significantly lower than in our setting; therefore, RCs represent a sporadic occurrence but still require prompt and effective management.

## 5. Conclusions

Respiratory patients’ management represents a daily clinical challenge in the heterogeneity of pediatric emergencies. Efficient triage systems and vital signs assessment are crucial for the prompt recognition and management of RCs. Vascular access, predominantly the peripheric one, was commonly established, with intraosseous access used in severe cases. Point-of-care tests provided valuable diagnostic insights. Despite the recognized utility of US, its utilization remained suboptimal in our setting. Collaborative efforts between ED and PICU are essential for improved outcomes. Our findings underscore the need for a systematic approach in pediatric emergency settings, supported by international and national guidelines but also by clearly defined diagnostic pathways, aiming to enhance the quality of care and patient outcomes.

## Figures and Tables

**Table 1 children-11-00462-t001:** Description of the five-level triage system, with the definitions and maximum waiting times of each level.

Triage Code	Denomination	Definition	Maximum Waiting Times
1—red	Emergency	Absence or compromission of one or more vital functions	Immediate evaluation
2—orange	Urgency	Stable condition with an evolutive risk of vital functions compromission or severe pain	Evaluation within 15 min
3—light blue	Deferred urgency	Stable condition without evolutionary risk but with implications for general health, typically requiring complex diagnostic–therapeutic interventions	Evaluation within 60 min
4—green	Minor urgency	Stable condition without evolutionary risk, typically requiring simple monospecialistic diagnostic–therapeutic interventions	Evaluation within 120 min
5—white	No urgency	Not urgent or of minimal clinical relevance	Evaluation within 240 min

**Table 2 children-11-00462-t002:** Criteria for defining pathologic items.

Item	Criteria for Pathologic Definition
Systolic blood pressure	<70 mmHg if age < 1 year
	OR
	<70 + (2 × age in years) if age > 1 year
	OR
	>the 90th percentile for age, gender, and height
Multiparametric blood gas analysis (MBG)	pH < 7.35 or >7.45
	OR
	blood glucose < 60 mg/dL or >200 mg/dL
	OR
	hemoglobin < anemia cutoff according to age
	OR
	serum sodium < 135 mmol/L or >145 mmol/L
Blood count	white blood cells < 5000/mm^3^ or >15,000/mm^3^
	OR
	platelets < 150,000/mm^3^
	OR
	hemoglobin < anemia cutoff according to age

**Table 3 children-11-00462-t003:** General descriptive characteristics of the total study population according to the medical and surgical pathways. Statistically significant differences (*p*-value < 0.05) between the medical and surgical populations are highlighted in bold fonts.

Variable	Total	Medical	Surgical
Sex, *n* (%)			
Male	522 (55.9%)	**438 **(**54.2%**)	**84 **(**66.7%**)
Female	412 (44.1%)	**370 **(**45.8%**)	**42 **(**33.3%**)
Age, median (percentile25-percentile75) years	3.2 (1.1–7.4)	2.7 (0.9–6.8)	6.6 (2.7–11.3)
Access mode to ED, *n* (%)			
Independently	414 (44.3%)	**404 **(**50%**)	**10 **(**7.9%**)
Ambulance	276 (29.6%)	**194 **(**24%**)	**82 **(**65.1%**)
Transfer from another hospital	189 (20.2%)	**157 **(**19.4%**)	**32 **(**25.4%**)
Not reported	55 (5.9%)	53 (6.6%)	2 (1.6%)
Main finding for red code definition, *n* (%)			
PAT	598 (64%)	**472 **(**58.4%**)	**126 **(**100%**)
SatO_2_	301 (32.2%)	**301 **(**37.2%**)	**0 **(**0%**)
HR	23 (2.5%)	**23 **(**2.8%**)	**0 **(**0%**)
RR	12 (1.3%)	**12 **(**1.6%**)	**0 **(**0%**)
Comorbidities, *n* (%)			
Yes	355 (38%)	**344 **(**42.6%**)	**11 **(**8.7%**)
No	577 (61.8%)	**463 **(**57.3%**)	**114 **(**90.5%**)
Not reported	2 (0.2%)	1 (0.1%)	1 (0.8%)

**Table 4 children-11-00462-t004:** Vital signs at ED admission of the study population according to the medical and surgical pathways. Statistically significant differences (*p*-value < 0.05) between medical and surgical populations are highlighted in bold fonts.

Variable	Total	Medical	Surgical
AVPU, *n* (%)			
A	598 (64%)	**538 **(**66.7%**)	**60 **(**47.6%**)
V	63 (6.8%)	**52 **(**6.4%**)	**11 **(**8.7%**)
P	119 (12.7%)	**103 **(**12.7%**)	**16 **(**12.7%**)
U	153 (16.4%)	**115 **(**14.2%**)	**38 **(**30.2%**)
Not reported	1 (0.1%)	0 (0%)	1 (0.8%)
Skin, *n* (%)			
Normal	598 (64%)	**578 **(**71.5%**)	**20 **(**15.9%**)
Pathologic	273 (29.2%)	**187 **(**23.2%**)	**86 **(**68.2%**)
Not reported	63 (6.8%)	43 (5.3%)	20 (15.9%)
Breathing, *n* (%)			
Spontaneous	718 (76.9%)	**633 **(**78.3%**)	**85 **(**67.5%**)
Oxygen support	159 (17%)	**152 **(**18.8%**)	**7 **(**5.5%**)
Intubation	56 (6%)	**23 **(**2.9%**)	**33 **(**26.2%**)
Not reported	1 (0.1%)	0 (0%)	1 (0.8%)
Blood pressure, *n* (%)			
Normal	162 (17.3%)	**109 **(**13.5%**)	**53 **(**42.1%**)
Pathologic	39 (4.2%)	**26 **(**3.2%**)	**13 **(**10.3%**)
Not reported	733 (78.5%)	673 (83.3%)	60 (47.6%)
Capillary refill time, *n* (%)			
Normal	233 (24.9%)	**224 **(**27.7%**)	**9 **(**7.1%**)
Pathologic	64 (6.9%)	**60 **(**7.4%**)	**4 **(**3.2%**)
Not reported	637 (68.2%)	524 (64.9%)	113 (89.7%)

**Table 5 children-11-00462-t005:** Clinical management of medical and surgical populations. Statistically significant differences (*p*-value < 0.05) between medical and surgical populations are highlighted in bold fonts.

Management	Medical	Surgical
Vascular access, *n* (%)		
PVC	**582 **(**72.1%**)	**110 **(**87.3%**)
IO	**5 **(**0.6%**)	**1 **(**0.7%**)
CVC	**1 **(**0.1%**)	**0 **(**0%**)
IO + PVC	**12 **(**1.4%**)	**3 **(**2.4%**)
No access	**202 **(**25.1%**)	**5 **(**4%**)
Blood count, *n* (%)		
Not performed	**254 **(**31.4%**)	**28 **(**22.2%**)
Performed	**554 **(**68.6%**)	**98 **(**77.8%**)
Pathologic	190 (34%)	49 (50%)
Blood gas analysis, *n* (%)		
Not performed	**172 **(**21.3%**)	**53 **(**42.1%**)
Performed	**636 **(**78.7%**)	**73 **(**57.9%**)
Pathologic	309 (48.6%)	20 (27%)
Electrocardiogram, *n* (%)		
Not performed	**637 **(**78.8%**)	**119 **(**94.4%**)
Performed	**171 **(**21.2%**)	**7 **(**5.6%**)
Pathologic	28 (16.4%)	2 (28.6%)
Ultrasound, *n* (%)		
Not performed	553 (68.4%)	80 (63.5%)
Performed	255 (31.6%)	46 (36.5%)
Pathologic	107 (42%)	13 (28.3%)
CT scan, *n* (%)		
Not performed	**726 **(**89.9%**)	**45 **(**35.7%**)
Performed	**82 **(**10.1%**)	**81 **(**64.3%**)
Pathologic	25 (30.5%)	55 (68%)
Anesthesiologic consultation, *n* (%)		
Yes	**206 **(**25.5%**)	**68 **(**54%**)
No	**602 **(**74.5%**)	**58 **(**46%**)

**Table 6 children-11-00462-t006:** Therapeutic approach in the total study population according to the medical and surgical pathways. Statistically significant differences (*p*-value < 0.05) between drug administration in the medical and surgical populations are highlighted in bold fonts.

Therapeutic Approach	Total	Medical	Surgical
Oxygen or ventilation, *n* (%)	548 (58.7%)	**521 **(**64.5%**)	**27 **(**21.4%**)
Crystalloids, *n* (%)	106 (11.3%)	**79 **(**9.8%**)	**27 **(**21.4%**)
Colloids, *n* (%)	28 (3%)	**20 **(**2.5%**)	**8 **(**6.3%**)
Glucagon or insulin, *n* (%)	26 (2.8%)	25 (3.1%)	1 (0.8%)
Electric therapy, *n* (%)	0 (0%)	0 (0%)	0 (0%)
Antiseizure, *n* (%)	98 (10.5%)	**96 **(**11.9%**)	**2 **(**1.6%**)
Sedatives, *n* (%)	47 (5%)	**26 **(**3.2%**)	**21 **(**16.7%**)
Psychoactive drugs, *n* (%)	11 (1.2%)	11 (1.4%)	0 (0%)
Antibiotics or antivirals, *n* (%)	172 (18.4%)	155 (19.2%)	17 (13.5%)
Antipyretics or analgesics, *n* (%)	269 (28.8%)	233 (28.8%)	36 (28.6%)
Antiemetics, *n* (%)	23 (2.5%)	17 (2.1%)	6 (4.8%)
Steroids, *n* (%)	260 (27.8%)	**257 **(**31.8%**)	**3 **(**2.4%**)
Drugs acting or airways, *n* (%)	376 (40.2%)	**376 **(**46.5%**)	**0 **(**0%**)
Diuretics, *n* (%)	12 (1.3%)	12 (1.5%)	0 (0%)
Cardioactive drugs, *n* (%)	27 (2.9%)	23 (2.8%)	4 (3.2%)
Dressings, *n* (%)	24 (2.6%)	**1 **(**0.1%**)	**23 **(**18.3%**)

**Table 7 children-11-00462-t007:** Hospital admission modality of the total study population according to the medical and surgical pathways. Statistically significant differences (*p*-value < 0.05) between medical and surgical populations are highlighted in bold fonts.

Admission Modality	Total	Medical	Surgical
Pediatric intensive care unit, *n* (%)			
No	759 (81.3%)	**688** (**85.1%**)	**71** (**56.3%**)
Yes	175 (18.7%)	**120** (**14.9%**)	**55** (**43.7%**)
Short observation unit, *n* (%)			
No	660 (70.7%)	**539** (**66.7%**)	**121** (**96%**)
Yes	274 (29.3%)	**269** (**33.3%**)	**5** (**4%**)
Hospital ward, *n* (%)			
No	232 (24.8%)	**216** (**26.7%**)	**16** (**12.7%**)
Yes	702 (75.2%)	**592** (**73.3%**)	**110** (**87.3%**)

**Table 8 children-11-00462-t008:** Mean duration of hospitalization according to the main complaint.

Main Complaint	Mean Duration of Hospitalization, Days (SD)
Respiratory	12 (±17)
Neuropsychiatric	11 (±16)
Traumatic	14 (±17)
Cardiologic	10 (±14)
Metabolic	7 (±4)
Surgical	18 (±17)

**Table 9 children-11-00462-t009:** Statistically significant correlations between diagnostic pathways and ED management items.

Pathway	Reason for RC Assignment	Characteristics at ED Admission	Clinical Management	Therapeutic Approach	Admission Modality	Deaths
Respiratory (438 patients, 46.9%)	SatO_2_	AVPU: A. B: oxygen. Normal skin.	MBG analysis: more than expected. Blood count, vascular access, and anesthesiologic consultation: less than expected.	Oxygen Steroids Drugs acting on airways Antibiotics and antivirals	PICU: less than expected. Short observation unit: more than expected.	Less than expected
Neuropsychiatric (249 patients, 26.7%)	PAT	Comorbidities. AVPU: P or U	Electrocardiogram: more than expected	Antiseizure Psychoactive drugs Antiemetics		
Traumatic (110 patients, 11.8%)	PAT	Transported by ambulance. No comorbidities. AVPU: U. B: intubated. Pathologic skin.	Blood count and anesthesiologic consultation: more than expected.	Crystalloids Colloids Sedatives Dressings	PICU: more than expected	More than expected
Cardiologic (87 patients, 9.3%)		Pathologic skin.	Electrocardiogram: more than expected	Crystalloids Colloids Cardioactive drugs		More than expected
Metabolic (36 patients, 3.8%)	PAT		MBG analysis and electrocardiogram: more than expected	Crystalloids Glucagon Insulin		
Surgical (14 patients, 1.5%)		Pathologic skin.	Blood count: more than expected	Dressings		

## Data Availability

The data presented in this study are available on request from the corresponding author due to the ongoing study.

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
