# Peer review of "Red Code Management in a Pediatric Emergency Department: A Retrospective Study"

_children, 2024, doi:10.3390/children11040462_

Round 1

Reviewer 1 Report

Comments and Suggestions for Authors

Thank you for the opportunity to read the interesting work on "red code" in children. I ask the authors to take the following comments into account before further processing the manuscript:

1) I don't understand why the abbreviation RD is used for "red code". Isn't RC better?

2) I suggest using the full abbreviation (POCUS) for point-of-care ultrasound. The authors separated POC and US, and this is not correct.

3) Please explain in more detail (in the methodology) the criteria for assigning the patient to the RED CODE group.

4) Table 8 is illegible. I propose to use the same criteria (in the table) for individual groups of patients and determine EXACTLY their significance coefficient

5) A description of the limitations of the study should be developed.

6) The conclusions do not directly refer to the results obtained. I ask the authors to modify their conclusions based on the results obtained. The work does not assess the QUALITY of medical procedures, but only the quantity of them. The authors basically described the patient's profile, not the effectiveness of selected procedures (e.g. POCUS). Please remember this so that the conclusions do not go too far.

7) As many as 13 out of 34 items of literature cannot be considered current (over 5 years old). Please take into account several current scientific reports, including:

a) Rzońca E, Świeżewski SP, Gałązkowski R, Bień A, Kosowski A, Leszczyński P, Rzońca P. Neonatal Transport in the Practice of the Crews of the Polish Medical Air Rescue: A Retrospective Analysis. Int. J. Environ. Res. Public Health 2020, 17, 705.

DOI: 10.3390/ijerph17030705

b) Horodkova Y. Current situation’s analysis of the problem in intensive care of complicated community-acquired pneumonias in children. Crit. Care Innov. 2020; 3(2): 29-42. DOI: 10.32114/CCI.2020.3.2.29.42

Author Response

1) We have accepted your correction and replaced the abbreviation RD with RC.
2) We have accepted your suggestion and used the full abbreviation POCUS.

3) In the section “materials and methods” we have added a new paragraph to explain the criteria for assigning the patient to the RC group.

4) We have modified the table showing the diagnostic pathways, hoping it is now more understandable and readable. In this table we have included only the items with statistical significance, p<0.05.   

5) In the section “materials and methods” we have added a description of the limitations of the study. –

6) We have modified the conclusions to make them more consistent with the content of the article.
7) We have considered and included the first article you suggested. As we mentioned in our manuscript, the international literature lacks studies specifically addressing red codes in pediatric age and closely related to our work. This is why we have also referenced articles that are not extremely recent.

Reviewer 2 Report

Comments and Suggestions for Authors

Thankyou for the opportunity to review your manuscript describing the characteristics of RED paediatric patients. 

It was an interesting read and contrast to the Australian Triage Scale and the paediatric setting adds another layer of complexity too. 

Overall it was a valuable retrospective review although there were some areas that could be expanded for context for international readers

Abstract - clear

Introduction.
The background should include the international perspective of how ED triage occurs eg in Australia they use numbers rather than colours. Its unclear if this has been taken into account when determining the literature review.

A table listing the triage scale and description would be useful to understand the context. 

There is discussion about 'our ED' - this should be moved to 2.1 and expanded - the ED in general needs to be described better ie where, how many presentations per year, supported by what kinds of hospital, size of catchment and other EDs close by that see children. 

The background needs to explain why the question you are asking is important and what you hope to achieve by answering it. This has not been covered at all. 

Materials/Methods

No ethical approval exists.
2.1 - see above for description of ED
2.1 - the triage evolution and description should be moved to the background

No description about inclusion/exclusion, date range of study etc

2.3 Pathologic criteria is not clear - does it inform the RED call? Given that blood gas analysis or blood count is one of them, it doesnt seem that it would work at triage?

Results

Comprehensive. Do you have the total number of presentations for that period too?

3.3 - was there differentiation between ambulance vs ED cannulation?

Do you have the data for ICD-10 codes for these patients as that would provide a way to compare and contrast with other EDs. 

Discussion

Postulation as to the reasons for 'not reported' access to the ED would be useful.

Im surprised at the low rate of PIVC cannulation and blood tests performed - would be good to discuss some of the reasons this may be.

No limitations mentioned

Comments on the Quality of English Language

a few small grammatical errors in sentence structure only

Author Response

Introduction
-We have added a table listing the triage to make more understandable the context. 
-We have moved the description of our ED POC tests to section 2.1, where we have also expanded the description of our hospital and our ED.  

-We have specified that in the international literature there is a scarcity of epidemiological studies related to RCs ED admissions in the pediatric age, so we aimed to provide a systematic approach in pediatric emergency setting. 

- we have added a paragraph reporting the description of our hospital, the annual ED accesses and the geographical context.

Materials/Methods
-as written in the “ethical declarations” section, The study was performed according to the international regulatory guidelines and current codes of Good Epidemiological Practice. In accordance with European regulations and laws, Italian observational studies from data obtained without any additional therapy or monitoring procedure do not need the approval of an independent ethical committee (https://www.gazzettaufficiale.it/eli/id/2003/08/09/003G0229/sg) (Italian)

- We have included ALL the patients triaged as red codes to our ED (there weren’t exclusion criteria) between 1 July 2020 and 30 June 2023. This time period is specified in the section “materials and methods”.

- In the section “materials and methods” we have added a new paragraph to explain the criteria for assigning the patient to the RC group. In the paragraph “pathologic items definition” we have specified the criteria we used to define pathologic items illustrated in table 4 and 5.

Results
-As illustrated in paragraph 3.1 Population study, we had 105.798 total ED admissions in the period of the analysis.

-As per your suggestion, we have specified that we have not make differentiation between ambulance vs ED cannulation.

-Unfortunately, we don’t have the data for ICD-10 codes because we haven’t collected them during the database making process.

Discussion
- In the section “materials and methods” we have added a description of the limitations of the study where we have specified there was a percentage of "not reported" data, either because they were not detected or simply not recorded in the clinical records.

-Regarding the low rate of PVC cannulation and blood tests performed, we suppose you are referring to the respiratory pathway and we have motivated  this aspect in our discussion with the high confidence of our ED team in managing respiratory RCs.

Round 2

Reviewer 1 Report

Comments and Suggestions for Authors

accept